

# Galapagos land iguanas as ecosystem engineers

Washington Tapia[1,2] and James P. Gibbs[2,3]

[1] Science Faculty, University of Malaga, Malaga, Spain
[2] Galapagos Conservancy, Fairfax, VA, United States of America
[3] Environmental Biology, State University of New York College of Environmental Science and Forestry, Syracuse, NY, United States of America

## ABSTRACT

**Background**. Declines of large-bodied herbivorous reptiles are well documented, but the consequences for ecosystem function are not. Understanding how large-bodied herbivorous reptiles engineer ecosystems is relevant given the current interest in restoration of tropical islands where extinction rates are disproportionately high and reptiles are prominent as herbivores.

**Methods**. In this study, we measured the ecosystem-level outcomes of long-term quasi-experiment represented by two adjacent islands within the Galapagos Archipelago, one with and the other without Galapagos land iguanas (*Conolophus subcristatus*), large-bodied herbivores known to feed on many plant species. We characterized plant communities on each island by developing high-resolution ($<1$ cm$^2$) aerial imagery and delineating extent of plant associations and counting individual plants on each.

**Results**. In the presence of iguanas there was dramatically less woody plant cover, more area with seasonal grasses, and many fewer cacti. Cacti had a more clumped distribution where iguanas were absent than where iguanas were present.

**Discussion**. This study provided strong evidence that Galapagos land iguanas can substantially engineer the structure of terrestrial plant communities; therefore, restoration of large-bodied reptilian herbivores, such as land iguanas and giant tortoises, should be regarded as an important component of overall ecosystem restoration, especially for tropical islands from which they have been extirpated.

## INTRODUCTION

Reptiles are prominent as herbivores in tropical island ecosystems and declines in their populations on islands have been particularly severe (*Gibbons et al., 2000*). Reptiles are thought to "engineer" island ecosystems through herbivory, seed dispersal and nutrient cycling (*Cooper Jr & Vitt, 2002*; *Falcón & Hansen, 2018*; *Valido & Olesen, 2019*) but the consequences of reptile declines for island ecosystems are not well known (*Malhi et al., 2016*; *Pérez-Méndez, Jordano & Valido, 2018*). The most definitive means of testing potential ecosystem-level effects of large-bodied reptiles is experimental manipulations of population numbers usually not possible because manipulation of entire landscapes is not

Corresponding author
James P. Gibbs, jpgibbs@esf.edu

feasible (*Debinski & Holt, 2000*). Quasi-experiments (*Shadish, Campbell & Cook, 2002*) do exist that have many of the qualities of designed, controlled experiments and take the form of ecosystem contrasts between adjacent islands with and without herbivores (*e.g.*, *Ali, 2004*), providing an opportunity to assess causal processes (*Beatty, Cox & Kuzee, 2018*).

In this study, we identified a fortuitous quasi-experimental situation represented by two adjacent islands within the Galapagos Archipelago, one with and the other without Galapagos land iguanas (*Conolophus subcristatus*, Fig. 1). This species along with two other species of land iguanas in Galapagos (*C. marthae* and *C. pallidus*) and giant tortoises (*Chelonoidis* spp.) once dominated as the only large-bodied herbivores present in the Galapagos Islands (*Fabiani et al., 2011*; *Tzika et al., 2008*). Today, all three species of land iguanas in Galapagos are listed as vulnerable or critically endangered due to past population collapses from predation by invasive predators, mainly dogs and humans, and habitat disruption by invasive herbivores, mainly goats (*Kumar, Gentile & Grant, 2020*).

Land iguanas are postulated to be ecosystem engineers for several reasons. They disperse seeds over large distances (*Traveset et al., 2016*) and also feed on many plant species (*Costantini et al., 2005*). These include grasses and herbaceous plants, as well as leaves and floral parts of woody plants, potentially affecting recruitment of woody plants and thereby mediating woody plant-grass interactions in the savannah-type ecosystems where land iguanas occur. *Opuntia* cactus is another important food source for land iguanas. In consuming fallen cactus pads and fruits, land iguanas might diminish asexual (vegetative) reproduction while enhancing sexual reproduction *via* seed dispersal away from adult plants where bird predation on seeds is intense (*Heleno et al., 2011*; *Nogales et al., 2017*). The cacti are, in turn, a keystone resource for much of the terrestrial vertebrate animal community (*Grant & Grant, 1981*).

To examine the ecosystem-scale impacts of land iguanas, we contrasted plant community composition on the two islands, which were similarly sized, immediately adjacent and comparable in most ways except for herbivore presence thereby largely controlling for factors potentially structuring plant communities unrelated to reptile herbivory. We characterized plant communities on each island by developing high-resolution (<1 cm$^2$) aerial imagery and delineating extent of plant associations on each. Our study provided an opportunity to ask questions about the role of reptilian herbivores on structuring the plant communities of islands, including impacts on keystone plants, as well as to explore the ramifications of restoring reptile populations on islands to promote ecosystem recovery (*Hansen et al., 2010*).

## MATERIALS & METHODS

The Galapagos Islands are a volcanic archipelago straddling the equator 1000 km west of continental Ecuador (Fig. 1A). Climate is unusually dry and cold for their equatorial position with average annual rainfall around 500 mm in coastal areas and temperatures varying annually between only 15 to 21 C. This study focused on the Plaza Islands, which are typical of the 128 small islands that comprise the archipelago (*Peck & Kukalová-Peck, 1990*). South Plaza (0034′56.3″S,9009′57.0″W, 12 ha) and North Plaza (0034′36″S,9009′32″W, 9

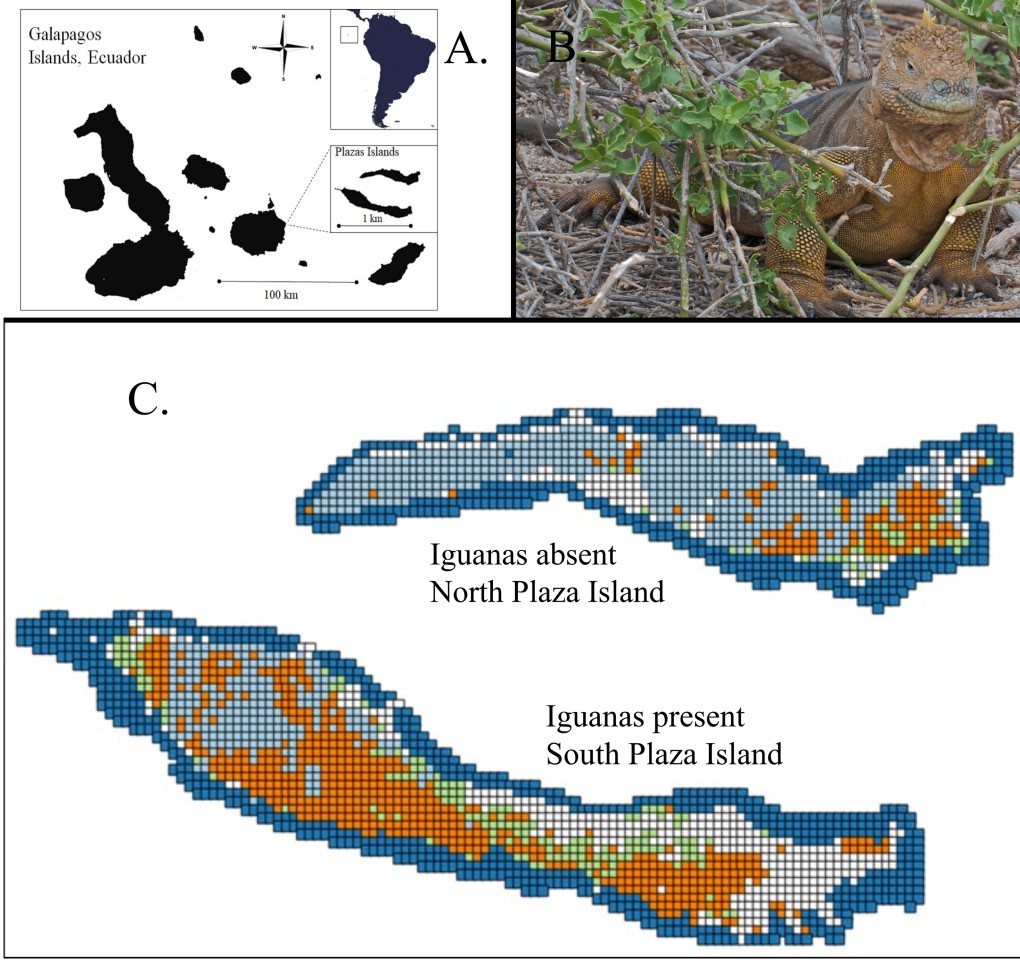

**Figure 1 The Plaza Islands in the Galapagos Archipelago, Ecuador where effects of land iguanas on terrestrial vegetation were assessed.** (A) Geographic location of Plaza Islands, Galapagos, Ecuador, (B) Galapagos land iguana (*Conolophus subcristatus*), South Plaza Island (image: A. Davey / Flickr, CC BY 2.0), (C) The North and South Plaza Island study sites. North Plaza Island lacked land iguanas whereas South Plaza Island supported iguanas (at a density of > 50/hectare). Maps of each island depict plant associations mapped from ultra-high resolution (1 cm² GSD) imagery obtained in 2012: woody plants (light blue), seasonal grasses (orange), succulents (light green), marine mammal impact areas (white), and intertidal zone (dark blue). Color of grid cell indicates the dominant vegetation type, *i.e.,* that with the highest proportion.

ha) are situated side-by-side 200 m apart and 500 m off the eastern coast of Santa Cruz Island (Figs. 1A, 1C). Each island has scattered expanses of soil on their uplands derived from the basaltic lava flows that comprise them. Plant communities are composed of shrubs, trees and tree-like cacti, and scattered seasonal grasses, herbs and sedges. Among 36 species of plants recorded on the Plaza islands, the following are the most widespread and form the basis for primary plant community associations: grasses and sedges—*Aristida subspicata* (Poaceae), *Bouteloua disticha* (Poaceae), *Cyperus anderssonii* (Cyperaceae), *Panicum laxum* (Poaceae), *Sporobolus pyramidatus* (Poaceae); woody plants—*Acacia rorudiana* (Mimosaceae),

*Bursera graveolens* (Burseraceae), *Maytenus octogona* (Celastraceae), *Parkinsonia aculeata* (Caesalpinaceae), and *Scutia spicata* (Rhamnaceae); succulents—*Sesuvium edmonstonei* (Aizoaceae), and cactus—*Opuntia echios* (Cactaceae).

The vertebrate fauna of the two islands differs primarily in terms of the presence and absence of land iguanas. Densities of land iguanas on South Plaza Island have been reported at >55 per hectare (*Snell & Christian, 1985*) whereas land iguanas have never been reported from North Plaza Island. Tourism is permitted on South Plaza Island and not on North Plaza Island but tourists are restricted to particular, narrow paths that generate little aggregate impact on the island's habitat (<1% trail-impacted). The elevation of the islands is similar but topography of South Plaza is more gradually sloping to its north shore whereas North Plaza is largely surrounded by cliffs that might permit less access to marine mammals (sea lions, *Zalophus wollebaeki*) to enter the island for resting. A final biotic difference known to occur between the islands other than presence/absence of land iguanas was the former existence of a small cohort of goats (five individuals were removed in 1961, *Campbell & Donlan, 2005*) and presence of house mice ca. 1982 to 2012 (now eradicated), both on South Plaza Island. During their period of occupation mice are hypothesized to have impacted cactus *via* burrowing into roots that in turn might have made cacti more vulnerable to toppling during wet periods (*Snell, Snell & Stone, 1994*) with iguanas simultaneously consuming cactus that might have fallen.

To obtain detailed data on vegetation for each island, we secured ultrahigh-resolution imagery of vegetation using an electric-powered, hand-launched octocopter of our own design (see *Fondriest, 2014*). Pre-planned fight paths were uploaded and autonomously executed by the aircraft at 50 m flight altitude over 8 h on May 12, 2012. The optical payload consisted of a commercial, off-the-shelf Canon S110 10 megapixel digital single-lens reflex camera suspended on a custom-designed, passive gimbal. Two images were captured per second with auto adjustment for exposure during each flight producing images with approximately 1 cm$^2$ ground resolution. Post-processing of the imagery was accomplished with Agisoft Photoscan image stitching software (Agisoft LLC, St. Petersburg, Russia) using the Geospatial Data Abstraction Library for geo encoding, warping and tiling supported with Python 3.4 and Imagemagick software (*ImageMagick Development Team, 2021*). Fieldwork was performed under research permit PC-82-14 granted by the Galapagos National Park Directorate.

Once the ultrahigh resolution imagery was mosaicked, we quantified vegetation on each island by overlaying a 5 × 5 m grid and estimating visually, based on expert knowledge of ground conditions, the proportion of each grid cell made up by the following: *Grasses*: seasonal grasses and herbaceous vegetation that otherwise were underlain by soil; *Succulents*: succulent plants that form dense monospecific mats over parts of these islands, *Woody plants*: shrubs or small trees; *Cactus*: the arboreal cactus present; and *Marine mammal impact areas*: areas with sparse plant growth distinguished by the bright white appearance resulting from accumulated feces being compacted and polished by mammals "hauling" over them repeatedly. We additionally counted the number of individual cactus plants occurring in each 5x5 m grid cell. To assess differences in overall plant community compositions between islands, we performed nonmetric multidimensional scaling using

Bray–Curtis dissimilarities between islands (nMDS(); vegan package) in R (*R Core Team, 2017*). To measure impacts of iguanas on cactus distribution, we assumed a Poisson process and indexed the spatial dispersion of cacti in the presence and absence of iguanas using the ratio of the variance-to-mean count of cacti (*Clapham, 1936*).

## RESULTS

We classified extent of vegetation on 1,148, 5x5 m grid cells on North Plaza Island (iguanas absent) and 1,777 grid cells on South Plaza island (iguanas present) (Fig. 1C). Vegetation composition differed substantially between islands: in the presence of iguanas there was less woody plant cover and more area with seasonal grasses (Fig. 2). Notably there was less cactus in the presence of iguanas (Fig. 2). Correlation of woody cover extent in each grid cell *versus* its extent in the eight neighboring grid cells (Fig. 2) indicated that cells with woody plant cover tended to be surrounded by more cells with a higher proportion of woody plants, on average, in the presence of iguanas than in their absence. The dispersion parameter for cactus was considerably lower (2.94) where iguanas were present, indicating a less clumped dispersion of cactus, than where iguanas were absent (4.05). Multivariate assessment of plant communities on each island (Fig. 3) revealed that areas dominated by grasses only and marine mammal impacts only were unique to the island with iguanas present whereas areas with 100% woody plant cover were unique to the island lacking iguanas. Large extents of both islands shared plant community associations characterized as combinations of woody plants and grasses as well as grasses and succulents, with some further overlap in areas dominated by marine mammal impacts and succulents.

## DISCUSSION

This study provides evidence that large-bodied, herbivorous reptiles can substantially engineer the structure of terrestrial plant communities. Not only was woody vegetation far less extensive on the island with land iguanas, the spatial pattern of woody vegetation also differed insofar as it tended to be surrounded by more woody vegetation in the presence of iguanas than in their absence. Land iguana impacts on cactus—a keystone species for the entire vertebrate community—were also substantial, reducing cactus abundance and altering spatial distribution of cactus.

We suspect the dramatic contrasts in vegetation between islands were due primarily to herbivory by iguanas. Land iguanas consume fruits, flowers, leaves and shoots of woody plants, including those of species that dominated on the Plazas Islands (*Christian, Tracy & Porter, 1984*; *Traveset et al., 2016*). Targeted herbivory on vulnerable parts (leaves, shoots) of regenerating woody plants is likely the primary mechanism by which land iguanas reduced woody plant cover because these large-bodied land iguanas, due to their weight (up to 10 kg), cannot be supported on peripheral branches of shrubs and trees and cannot consume their crown foliage. As for cactus, land iguanas primarily consume fallen cladodes, but also consume fallen fruits. Land iguana consumption of fallen cladodes near adults eliminates asexual reproduction in cacti that generates clustered distribution of cacti, whereas iguanas dispersing seeds away from adult cacti and the intense seed predation

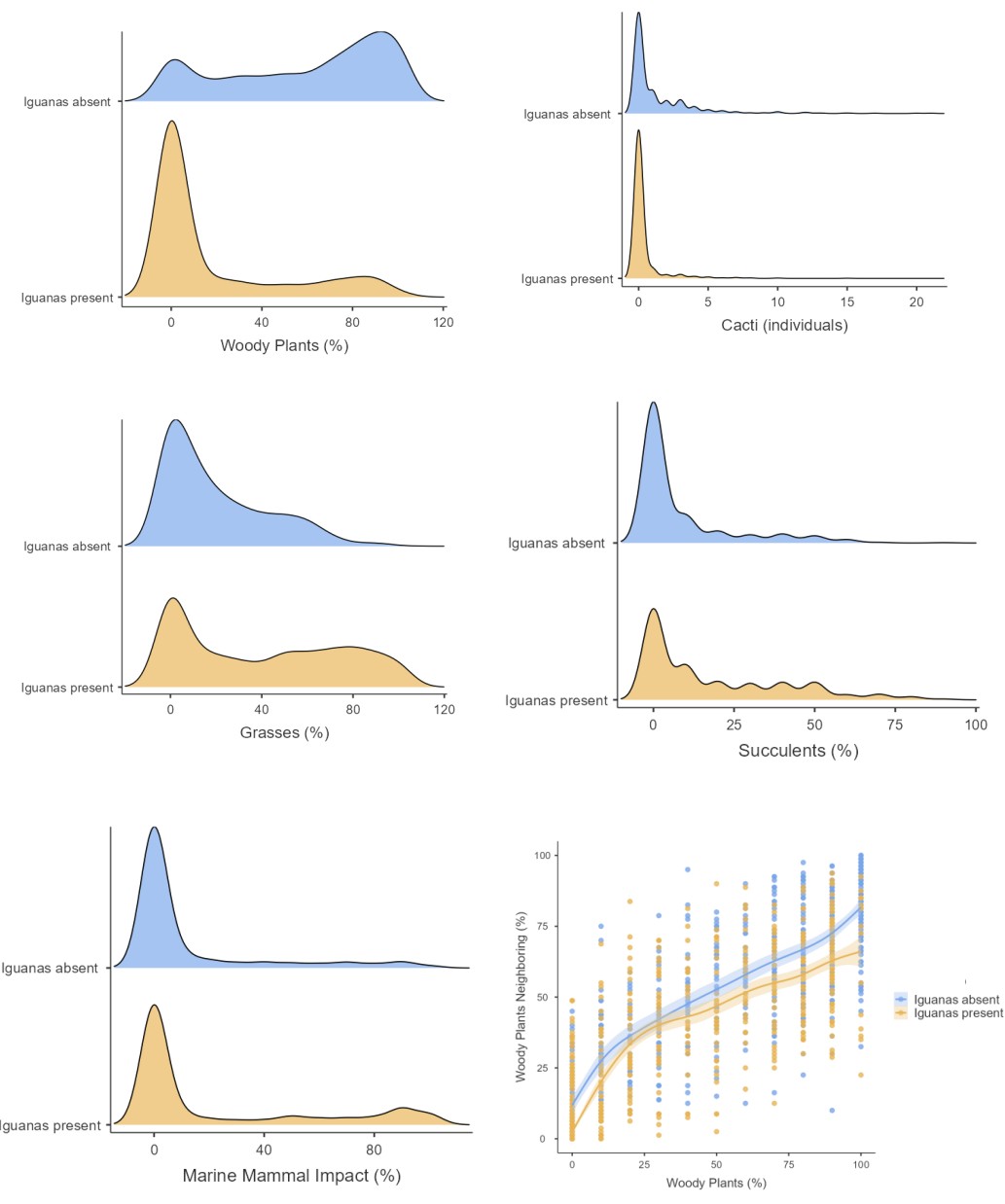

**Figure 2** **Contrasts between plant community parameters on two adjacent islands in the Galapagos, one with iguanas present and one with iguanas absent.** Contrasts between plant community parameters on two adjacent islands in the Galapagos, one with iguanas present and one with iguanas absent. All plots depict the probability densities of extent of different vegetation types on North Plaza Island (iguanas absent; 1,148, 5 × 5 m grid cells) and South Plaza Island (iguanas present; 1,777 grid cells), except plot on lower right, which depicts woody cover extent in each 5 × 5 m grid cell *versus* extent of woody vegetation in the eight neighboring grid cells on each island Intervals (LOESS smoothed and bounded by 95% confidence intervals).

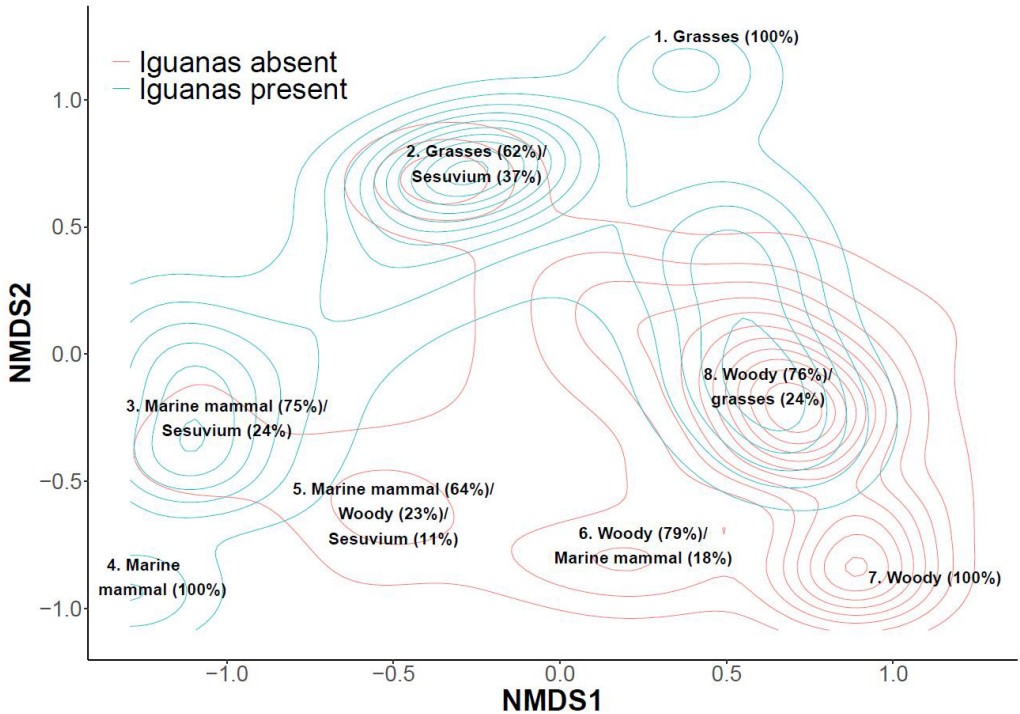

**Figure 3** Nonmetric multidimensional scaling plot visualizing differences in plant community associations on two adjacent islands in the Galapagos Archipelago, one with land iguanas present and one with land iguanas absent. Ordination based on Bray–Curtis dissimilarity index is split by island: iguanas present (South Plaza) and absent (North Plaza). The plot presents the kernel density estimation of the distribution of 1,148, 5 × 5 m grid cells on the island with iguanas absent and 1,777 5 × 5 m grid cells the island with iguanas present (derived from kde2d(); MASS package). To aid in interpretation, mean values for each univariate vegetation parameter comprising > 10% of the cells at a given peak are presented for each of the eight point clusters identified.

that occurs there by birds (*Nogales et al., 2017*), thereby promoting sexual reproduction in cactus and wider dispersion of cactus individuals, a pattern we observed.

Land iguana herbivory might well have a cascading effect on the biotic community of small oceanic islands, by influencing other terrestrial vertebrates through changes in habitat structure and composition. One important interaction likely resulting from iguana herbivory with impacts on many other species on the Plazas Islands involves marine mammals. Sea lions cannot navigate through woody vegetation when seeking basking sites, and occupy more sparsely vegetated areas facilitated by iguanas. Sea lions deposit prolific amounts of guano with attendant changes in the soil chemistry (*Fariña et al., 2003*). Notably the extent of marine mammal impacts was greater on the iguana-occupied island (Fig. 1C). Further studies contrasting the animal communities (birds, other lizards, invertebrates) would inform how iguana-triggered changes in plant communities trigger cascading into the larger biotic community on these islands.

A primary limitation of our study is that it represents a pseudo-replicated design with one replicate within each treatment. Unfortunately given the widespread extinctions of island forms of large-bodied reptilian herbivores elsewhere in Galapagos and around

the world (*Simberloff, 1976*; *Slavenko et al., 2016*; *Foufopoulos & Ives, 1999*), there are few opportunities to increase the replicate number (number of islands) to strengthen inference. The historical occurrences of a tiny population of goats (1960′s) as well as house mice (1-2 decades previously) on South Plaza Island might have impacted vegetation at the time of their tenure (*Campbell, Carrión, & Sevilla, 2011*; *Snell & Christian, 1985*) but are unlikely to account for the striking differences in plant communities evident today. A significant structuring agent, however, might be marine mammals. South Plaza presents a more gradual northern slope such that it is more heavily accessed by sea lions for resting. Land iguanas might facilitate sea lion use of the island's uplands by reducing woody plant cover, with sea lions then *via* trampling and altering soil chemistry through deposition of feces sea lions affecting vegetation. This said, many parts of South Plaza Island remain inaccessible to sea lions and those areas still evidence the general differences observed in vegetation (reduced woody plant cover, lack of cactus) between North and South Plazas Islands.

Why land iguanas do not occur on North Plaza Island is not clear. The species' habitat is characterized as ''dry areas with low growing shrubs and Opuntia cactus'' (*Kumar, Gentile & Grant, 2020*) which describes well North Plaza Island. Indeed, the abundance of cactus, woody plants and expanses of grasses on North Plaza Island suggests the habitat might be of high quality for land iguanas. Nesting habitat for land iguanas also is not likely limiting on North Plaza Island given the expanse of open soil present. We expect that land iguanas either never succeeded in colonizing this small offshore island (*Hedrick, 2019*) despite being potentially able to do so, or that they did colonize it but went extinct historically.

## CONCLUSIONS

Understanding how large-bodied herbivores engineer ecosystems is relevant today given widespread, current interest in reintroducing extant species back to places from which they were extirpated in historical times (*Seddon, 2010*; *Johnson et al., 2018*). With many proposed and in some cases ongoing trophic rewilding programs involving reptiles on islands premised upon the largely unevaluated assertion that restoring reptiles to islands will re-instate key ecological functions (*Frazier, 2021*), a better understanding of herbivore impacts on island ecosystems is required. An important outcome of this study is highlighting the importance of top-down effects of reptile herbivores and the potential of herbivore restoration to facilitate ecosystem recovery. We tested the hypothesis that plant communities are differently structured in the presence *versus* absence of land iguanas. We provide evidence that land iguanas substantially engineer the structure of plant communities. This study suggests that the widespread extinction of reptile herbivores, which once served as the dominant herbivore in many tropical oceanic ecosystems, might have profound implications for the status of these ecosystems, and the species that comprise them, today. Restoration of large-bodied reptilian herbivores, such as land iguanas and giant tortoises, should be regarded as an important component of restoration in ecosystems where they have been extirpated.

## ACKNOWLEDGEMENTS

We are grateful to Sean Burnett and Greg Carney for their efforts to secure and process the aerial imagery, to Sahila Kuldalkar for assistance in classifying the aerial imagery, and to the Galapagos National Park Directorate for permission and support to image the study sites. The manuscript was greatly improved thanks to the comments of three reviewers.

### Funding

This work was supported by the Galapagos Conservancy and the Galapagos National Park Directorate. The funders had no role in study design, data collection and analysis, decision to publish, or preparation of the manuscript.

### Grant Disclosures

The following grant information was disclosed by the authors:
The Galapagos Conservancy.
The Galapagos National Park Directorate.

### Competing Interests

The authors declare there are no competing interests.

### Author Contributions

- Washington Tapia conceived and designed the experiments, performed the experiments, authored or reviewed drafts of the paper, logistics and management of research, and approved the final draft.
- James P. Gibbs conceived and designed the experiments, performed the experiments, analyzed the data, prepared figures and/or tables, authored or reviewed drafts of the paper, and approved the final draft.

### Field Study Permissions

The following information was supplied relating to field study approvals (i.e., approving body and any reference numbers):

This research was conducted under a research permit granted by the Galapagos National Park Directorate to perform fieldwork on the two Plazas Islands (PC-82-14).

### Data Availability

The raw data are available in the Supplementary File.

### Supplemental Information

Supplemental information for this article can be found online at http://dx.doi.org/10.7717/peerj.12711#supplemental-information.

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
