# Peer review of "Galapagos land iguanas as ecosystem engineers"

_PeerJ, doi:10.7717/peerj.12711_

## Round 0.1 · original submission · Major Revisions

Thank you for your submission to PeerJ. Three expert reviewers have reviewed this manuscript and have provided thoughtful and constructive feedback. While overall the manuscript is well written and clear, the reviewers make some important points about the language used. Is "rewilding" really the most appropriate term to be used here? Can you be confident given the study design that the lizards are engineering plant communities, as implied in the manuscript? Or are plant communities simply correlated with lizard presence?

Please take each reviewers suggestions into account when revising your manuscript. I look forward to reading it!

Reviewer 1 ·

Basic reporting

The writing is generally clear, but some sentences area a little convoluted and long. For example:
- ln 42: "a process envisioned with contemporary ecological theory built around interaction networks" - this phrase is likely not needed.
- ln 43-46: This sentence is too long. Why is it difficult to estimate the "magnitude of effect of one species on the distributions and abundances of others"?

Literature references are generally good, but there were a few spots where a connection to the literature could have been improved.
- ln 62: "impacts of reptiles on vegetation are essentially unknown." There is a wealth of knowledge on the impacts of the other large reptilian herbivore in the Galapagos, giant tortoises. There are many papers that could be cited, including those by the authors and highlighted in their recent book. In addition, there are many papers on effects of Aldabran tortoises and other large tortoises around the world on plant communities. This phrase should be removed.
- ln 71: The citation to Merlin & Juvik 1992 is about a quasi-experiment with introduced herbivores. I would appreciate seeing an example (or justification) of a quasi-experiment with naturally occurring herbivores that may have been present for long periods of time (see my concern about the study design and inference below). The Beatty et al. citation does not seem to be about quasi-experiments with herbivores.

I appreciate the sharing of the raw data, but it could use a metadata description of what the column headers mean.

Experimental design

It is not clear to me why there is a focus on "rewilding" in the title and introduction of the paper. There is not any translocation being conducted in the study, it is more about the ecosystem impacts of herbivores. There are implications for the practice of rewilding, but since the herbivore was not manipulated (i.e. removed or introduced) it is a bit of a weak connection. I would suggest only bringing up rewilding in the discussion as it is an implication of the study (i.e., if iguanas have a large effect on their ecosystems, then there is an argument to be made that they need to be reintroduced to the ecosystems they once inhabited). With this in mind, it would also be helpful to know where land iguanas are currently distributed compared to where they were in the past. It is not clear from the introduction that land iguanas are even in need of reintroduction. Given statements such as "land iguanas often dominate as the only large-bodied herbivore present" (line 76) - this seems to indicate that they are present everywhere that they should be, and thus not in need of reintroduction. (Also, what about giant tortoises as large-bodied herbivores?)

It would be good to see some hypotheses/predictions in the introduction about what you expected to find. There are some hints at this (e.g., line 81, where there is a discussion of iguana effect on cactus reproductive mode), but this could be made clearer as a specific prediction.

Methods:
There is a good description of the two islands, but there are two key differences about the islands that are not mentioned, both of which could potentially affect the plant community. South Plaza Island is a tourist site with high visitation rates; whereas North Plaza is prohibited from visitation. Although tourists are instructed to stay on trails, there is inevitably some wandering (and thus trampling) that could influence the vegetation. South Plaza Island also has a substantially steeper slope than North Plaza, which can influence water flow/infiltration and thus what plants are present. These differences need to be described as they could both have impacts on plant community composition.
Analysis:
It is not clear what the "experimental unit" is in the Bayesian ANOVAs to estimate the effect of treatment. (It is also not clear why this is a Bayesian analysis, when a standard ANOVA would seem to be suitable for a one factor analysis.) Arguably, there is a single replicate for each treatment (each island is a replicate, as the authors acknowledge in the discussion), but that is clearly not the approach taken in the analysis as statistics could not be estimated with a single replicate. It seems from the results that each 5x5m grid cell is likely taken to be an experimental unit, resulting in the extremely high and extremely low values in Table 1. I don't think it's appropriate to consider each grid cell to be an independent unit - there is a lot of spatial autocorrelation in vegetation cover, and some of these individual plants may cover more than one 5x5m cell. I see 2 options: 1) remove the Bayesian ANOVA analysis and simply report the %cover differences and nMDS analysis which are convincing enough of the differences between the islands, or 2) account for spatial autocorrelation in the statistical analysis. There are potentially multiple ways to do this, an easy thing to do would be to subsample the pixels at each island to reduce the chances that a sampled 5x5m cell is near another sampled cell.

Table 1: More description of what the Bayes Factors mean would be helpful, as this is not a common analysis. What does a large BF10 indicate? Is there a threshold for interpretation?

Validity of the findings

The results are fairly clear in terms of the vegetation differences between the two islands. My concern is with the interpretation that these differences are entirely due to the effect of iguanas on the plant community, when it is entirely possible that the differences simply indicate what is habitat for the species. In any other system, if researchers described the plant community of where an animal is present and where it is absent, we would infer that the plant community characteristics of where it is present are descriptive of the species' habitat. The difference here is that this is an island system, and so the assumption is that iguanas cannot get to North Plaza. However, both of these islands are very close to the larger island of Santa Cruz and they are very close to each other. It seems very likely that, over evolutionary time, land iguanas would have gotten to North Plaza multiple times, and so the question is, why are they not there now? One potential explanation is that the conditions do not provide suitable habitat - perhaps woody plant density is too high, or grass cover too low. The authors argue that iguanas created the opposite conditions on South Plaza, but the converse could be equally true (that iguanas are only present because the conditions on South Plaza favor their presence). Without a true experiment, it is impossible to say whether the iguanas are present because of the habitat or that the iguanas are causing the plant community to be structured as it is. The authors need to at least discuss this possibility. It would be helpful if there was some discussion of the habitats that the species is found in throughout the archipelago.

Line 191: The sentence ends by stating that iguanas are targeting "woody plant regeneration" with their herbivory, but then the next sentence goes on to discuss cladodes, which I assume is referring to the cactus. There is no evidence provided that iguanas eat woody plants - either adults or saplings.

Generally, the discussion was a little superficial and did not delve into the results. Many questions remain unaddressed.
- There were substantially more marine mammal impacts on South Plaza (iguanas present) - could this be due to differences in topography? Could marine mammals be having an influence on the plant community in the rest of the island? The authors indicate that marine mammals deposit large amounts of guano (nitrogen) - how could this influence plants on the island?
- What would the mechanism be for how iguanas would promote grasses over woody plants? Giant tortoises might do this through trampling, but it is hard to see how iguanas would have the same impact.
- Please discuss more the potential impacts of the introduced mice. There is some thinking that the mice had a large impact on the regeneration of cactus (through eating seeds) - how might this play into the dispersion pattern that was reported on?

·

Basic reporting

In terms of basic reporting, the language is pitched at the right level, the manuscript is well-paced and well-structured, and the language is clear. The context is sufficient, except for the issues I have raised below. The figures are presented in a professional manner. The raw data has been shared and is well-described. The study is self-contained and the results are directly related to the hypothesis.

Comments on figures:
Figure 1 caption: Does the colour of the grid cell indicate the dominant vegetation type, i.e. the one with the highest proportion? If so then the caption should explicitly state this
Figure 2: The caption does not include a description of the top right panel which, in the text is referred to as “2b”.
Figure 2: The data provided with the manuscript has a precision of 10% for vegetation cover. This could be more faithfully represented graphically by using histograms with 11 bins. The kernel density estimates in Figure 2 imply higher resolution than the original data and, if used, require justification.

Comments on citations:
The citations do not clearly support the claims in several places
Lines 43-46: The claim that the effects of herbivory are difficult to predict is puzzling and this claim is a little vague. I suggest simply claiming that there is good evidence that biotic factors influence distribution and abundance (e.g. Wisz et al) and that herbivores often act in several ways to structure vegetation communities (e.g. Maron and Crone 2006) but that effects depend on the particulars of ecosystems.
Line 51: I suggest instead finding evidence of increasing numbers of rewilding projects, or cite similar claims by authorities (e.g. Svenning et al. 2016)
Lines 55-56: Islands are arguably not a major focus of rewilding, given the number of projects in mainland Europe. It would make more sense if the authors claimed that islands should be a major focus of herbivore rewilding for ecosystem recovery—a claim that is their own and needs no citation—and then continue with the very reasonable justification of that claim in the sentences that follow.
Line 69: The description of the study as a ‘quasi-experiment’ should be linked to Shadish et al 2002 rather than Allen et al 2017.


Typographic and grammatical errors:
Line 34: author order is incorrect
Line 35: suggest “…especially for tropical islands from which they have been extirpated…”
Line 53: suggest “…in the absence of herbivory”
Line 63: There is no need for quotation marks around the word ‘rewilding’
Line 83: suggest “…is intense”
Line 86: suggest ‘…iguanas, we contrasted’
Line 98: suggest ‘…Plaza islands”
Lines 105: suggest ‘…Plaza islands”
Line 121: suggest “…imagery”
Lines 133: suggest “…identify individual”
Lines 134: suggest “…delineate the extent”
Line 137: suggest “…cell made up by the following”
Line 143: suggest “…counted the number”



Maron, John L., and Elizabeth Crone. "Herbivory: effects on plant abundance, distribution and population growth." Proceedings of the Royal Society B: Biological Sciences 273.1601 (2006): 2575-2584.
Svenning, Jens-Christian, et al. "Science for a wilder Anthropocene: Synthesis and future directions for trophic rewilding research." Proceedings of the National Academy of Sciences 113.4 (2016): 898-906.
Wisz, Mary Susanne, et al. "The role of biotic interactions in shaping distributions and realised assemblages of species: implications for species distribution modelling." Biological reviews 88.1 (2013): 15-30.

Experimental design

The research is original, and within the scope of the journal. The investigation appears rigorous and the methods are, for the most part, described in full.

Comments on methods:
Line 151: I am not familiar with Bayesian ANOVA, so my comment may be overly cautious: The authors should show that the transformed data adequately meets the assumptions of Bayesian ANOVA. Certainly this would be the case for frequentist ANOVA (and even with an arcsin tranformation, in the latter case, the data would likely not meet those assumptions given the high proportion of cells with value “0”). Alternatively, the authors might consider a logit transformation and generalised linear regression (Warton and Hui 2011).

Lines 159: I did not understand why confidence intervals were generated. It was implied that the authors counted every cactus in every 5x5m grid cell, in which case they have the population mean and variance for the number of cactus per cell for each island. If the CIs were generated for some other reason, perhaps it should be stated.


Warton, David I., and Francis KC Hui. "The arcsine is asinine: the analysis of proportions in ecology." Ecology 92.1 (2011): 3-10.

Validity of the findings

The authors should more explicitly address the competing hypotheses for the underlying causes of differences in vegetation between the islands (see lines 87-89). To convince the reader that iguana feeding behaviour is the most likely cause of vegetation differences then the authors need to also convince the reader that the other plausible explanations are less likely. They can do this by giving more detailed evidence about
- the likelihood of lasting impacts of house mice
- the likelihood of lasting impacts of goats (e.g. see Campbell and Donlan 2005)
- differences between the island soils, topography and weather
- the effect of vegetation on sea lion basking sites and vice versa (to rule out the effect of sea lions on vegetation) (i.e. lines 202-204), if any observations have been published.
This extra information will make the claims better justified and make an already-interesting study more so.

The tone of the title, abstract and discussion promises somewhat more than the evidence delivers. Arguably, the authors did not 'demonstrate' (line 183) that large-bodied, herbivorous reptiles can substantially engineer the structure of terrestrial plant communities. They did, on the other hand, provide good evidence for that hypothesis. I suggest re-wording the claim to better reflect the limitations of the study.

Comments on results:
Line 170: do the authors mean “surrounded by more woody plants cells” rather than “surrounded by more woody plants”?
Lines 176-177: regarding the phrase “areas dominated by grasses only and marine mammal impacts only were unique to the island with iguanas present whereas areas dominated solely by woody plants were unique to the island lacking iguanas”, this should be re-stated for clarity. There are grass-dominated areas on both islands, for example (Figure 1). I suggest “cells with 100% woody plant cover” etc.

Minor comments on discussion:
Line 224-5: since this study is not about a recovering ecosystem, I suggest rewording this phrase to highlight instead the potential for restoring important ecosystem
Line 227: suggest “we provide some evidence that land iguanas substantially engineer …”
Line 232: suggest “restoration in systems where they have been extirpated”


Campbell, Karl, and C. Josh Donlan. "Feral goat eradications on islands." Conservation biology 19.5 (2005): 1362-1374.

Additional comments

Congratulations on taking advantage of a neat natural experiment and applying non-invasive methods to describe a striking comparison. It may have important implications for the trophic ecology and the restoration of island ecosystems.

Overall, I felt that the paper was well balanced and the methods and results were well described. However, the strength of the evidence for the hypothesis that iguana herbivory is the primary cause of vegetative difference is somewhat overstated in places. This is not to say that the results are not valuable and worth reporting—they are—but more a cautious conclusion and more evidence against competing conclusions will make the manuscript both more robust and more interesting.

·

Basic reporting

The present study aims to analyse the ecosystem-scale impacts of Galapagos land iguana (Conolophus subcristatus) on structuring plant communities. For this, authors compare plant communities (by using high-resolution maps) in two adjacent (and comparative) islands with/without iguanas to assess the role of this large-bodied herbivorous species on vegetation. They found that in presence of iguanas there was 208% less woody plant cover, 52% more area
with grasses, and 258% less cactus.

The topic of this MS is really interesting. However, some aspects of the MS (see below) need to be considered in order to improve it. My principal concern is related to the presence of goats (and house mice) in the island with Conolophus (South Plaza island). Thus, since iguanas and goats coincide recently in the past in this island, the role of goats on vegetation in South Island need to be discuss in more detail. Besides, instead these islands are very near each other and its expected similar levels of e.g. precipitation, I am wondering about other environmental factors which can affect vegetation cover instead presence of iguanas. I mean, e.g. soil/rock cover, soil type, soil nutrients, vegetation impact of human near in the past, presence of seabirds and others (because presence of guano), etc. These aspects need to be included in some way to know the real effects of iguanas on vegetation cover. Besides, the role of iguana such as herbivorous vs. seed dispersers need to be analysed in more detail. What about data of iguana’s diet in this specific island? Could authors also used historical aerial photos of these islands in those years with goats and house mice were eradicated? Could Opuntia species also being negatively affected by other biological factor instead Conolophus, e.g. some Coccoidea species?

Experimental design

no comment

Validity of the findings

no comment

---

## Round 0.2 · Minor Revisions

Thank you for submitting this revised manuscript. The manuscript has been reviewed by the three assessors and all have commented that the manuscript has improved.

Please address these reviewer comments in your revision, paying special attention to the suggestions made by Reviewer 3. It would be good to include soil and climatic data, and also to include a substantial consideration of other herbivores that might influence vegetation on the island, such as goats, rodents, sea lions.

Figure 2 is a little unclear and to me, my takeaway is that there is not much difference between the two groups across the habitats. Can you include central tendency and perhaps also overlay the data points? this will help the reader understand the spread a little better. Violin plots are fine, but sometimes boxplots are a little clearer (especially with the data points overlaid).

Thank you, I look forward to receiving your revised manuscript.

Reviewer 1 ·

Basic reporting

No comment

Experimental design

No Comment

Validity of the findings

No comment

Additional comments

The authors have done a good job responding to my comments and those of the other reviewers. I think that the paper is now ready for publication and will represent a meaningful contribution to the literature on reptile ecosystem engineering.

·

Basic reporting

Line 43: there seems to be some confusion around the Cook et al reference. The version I can see online is here: https://www.alnap.org/system/files/content/resource/files/main/147.pdf, which has the author order as Shadish, Cook, Campbell. Google Scholar has the author order “TD Cook, DT Campbell, W Shadish”. I’ll leave it to the copy editor to make a call on that one.
Line 45: suggest change ‘a’ to ‘an’
Line 145: suggest change ‘cactus considerably’ to ‘cactus was considerably’
Lines 207-208: The authors might consider that iguanas may have also colonised the island but lost the small population lottery
Line 226: ‘substantially’ is a repeated word
Line 226: ‘that, the’: suggest remove comma
Figure 2: the top right sub-figure does not match the description. Other parts of figure 2 are violin plots but this is a line plot. It is not clear from the caption why this figure differs from the others, and what it represents. In the text at lines 141-2 it is described thus: “Correlation of woody cover extent in each 142 5 x 5 m grid cell and extent of woody vegetation in the eight neighbouring grid cells (Fig. 2b)” but the subfigure itself is not marked ‘b’. I suggest marking the individual sub-figures and making the description explicit.

Experimental design

No comments

Validity of the findings

This version is argued in a more balanced and convincing way.
Lines 155, 162. If the evidence is ‘strong’ then the authors can do more than merely ‘suspect’ the effects were due to iguanas. I suggest ‘the evidence gives us good reason to believe that the dramatic effects… etc etc.’

Additional comments

I’m looking forward to seeing this in print.

·

Basic reporting

In this revised version the MS has been improved in relation to comments of previous reviewers. The topic (the impact of herbivores on insular vegetation) is really interesting. However, some questions (and doubts) to formulate that vegetation changes among these island can be entirely due to the effect of land iguanas are not well resolved yet. Below I have included some comments to be considered in this revised version:

L39.- Maybe authors can include other general references about reptiles such as herbivore and seed dispersers (instead only tortoises) to include herbivore (Cooper & Vitt 2002, J Zool), and seed disperser (Valido & Olesen 2019, Frontiers Ecol Evol) lizards, as wells.

L40.- Other examples about consequences of reptile declines on insular vegetation are the next: Perez-Mendez et al. 2016 (Sci Reports), Perez-Mendez et al. 2018 (J Ecol).

L52.- Include latin names of three species of Galapago land iguanas

L77. Mat & Methods. I also miss a detailed description of climate and soil conditions in both islands. These aspects can also affect vegetation cover and composition

L93.- Include “vertebrate fauna”

L155.- Since South Island include in some way feral goats, tourist, mice, sea lions, instead Conolophus, authors need to discuss in detail their specific impact on vegetation. I mean, can these factors affect the observed differences on vegetation cover, presence of grasses, and density of Opuntia? These aspects (such as alternative hypotheses) need to discuss in more detail at time to use the sentence “This study provide strong evidence…” I’ll see that previous reviewers asking this point too, but evidences about in the MS is not really convincing yet.

L193.- Relate to previous point (L331) related to presence of goats and mice, authors need to discuss in more detail this sentence “but are unlikely to account for the striking differences in plant communities evident today”. It is not clear that between island differences can be entirely due to the effect of land iguanas.

L198-201.- To support this conclusion, it would be interesting to see these data on Results, as wells.

In general the Discussion need to be re-written according to other potential factors influencing vegetation changes and showing clear evidences that they are less likely to compared with consequences of land iguanas.

Experimental design

'no comment'

Validity of the findings

'not comment'

Additional comments

'not comments'

---

## Round 0.3 · accepted · Accept

Thank you for your revision, you did an excellent job of addressing the reviewer comments. I have decided to accept the manuscript for publication, contingent on a few minor changes to wording and clarity. These line numbers are in reference to the track-changes version of the manuscript.

Line 25-26: I don’t think you can have >100% less than something… Perhaps this effect size is actually the other way, such that “when iguanas were not present, there was 208% more woody plant cover …” etc? Please double check the change in effect is presented in the correct way (I always confuse this myself and just want to make sure the effect sizes are stated such that they reflect the data accurately).

Line 74 – replace “may” with “might”, here and throughout to be grammatically correct
Line 79 – the newly added “which” should be “that”

Line 97 – there is an extra “ .”
Line 167 – “indicated” is more appropriate than “suggested” here and throughout – because data/results indicate while people suggest

Lines 226-237 – This sentence is a little confusing, with a misplaced modifier. Think about rephrasing for clarity

Lines 255-258 – This is a long sentence and a little difficult to understand. Rephrase for ease and clarity